# Pt-Based Oxygen Reduction Reaction Catalysts in Proton Exchange Membrane Fuel Cells: Controllable Preparation and Structural Design of Catalytic Layer

**DOI:** 10.3390/nano12234173

**Published:** 2022-11-24

**Authors:** Hongda Li, Hao Zhao, Boran Tao, Guoxiao Xu, Shaonan Gu, Guofu Wang, Haixin Chang

**Affiliations:** 1Liuzhou Key Laboratory for New Energy Vehicle Power Lithium Battery, School of Electronic Engineering, Guangxi University of Science and Technology, Liuzhou 545006, China; 2Quantum-Nano Matter and Device Lab, State Key Laboratory of Material Processing and Die & Mould Technology, School of Materials Science and Engineering, Huazhong University of Science and Technology, Wuhan 430074, China; 3Key Laboratory of Fine Chemicals in Universities of Shandong, Jinan Engineering Laboratory for Multi-Scale Functional Materials, School of Chemistry and Chemical Engineering, Qilu University of Technology (Shandong Academy of Sciences), Jinan 250353, China

**Keywords:** Pt-based catalysts, oxygen reduction reaction (ORR), catalytic layer, proton exchange membrane fuel cell (PEMFC), membrane electrode assembly (MEA)

## Abstract

Proton exchange membrane fuel cells (PEMFCs) have attracted extensive attention because of their high efficiency, environmental friendliness, and lack of noise pollution. However, PEMFCs still face many difficulties in practical application, such as insufficient power density, high cost, and poor durability. The main reason for these difficulties is the slow oxygen reduction reaction (ORR) on the cathode due to the insufficient stability and catalytic activity of the catalyst. Therefore, it is very important to develop advanced platinum (Pt)-based catalysts to realize low Pt loads and long-term operation of membrane electrode assembly (MEA) modules to improve the performance of PEMFC. At present, the research on PEMFC has mainly been focused on two areas: Pt-based catalysts and the structural design of catalytic layers. This review focused on the latest research progress of the controllable preparation of Pt-based ORR catalysts and structural design of catalytic layers in PEMFC. Firstly, the design principle of advanced Pt-based catalysts was introduced. Secondly, the controllable preparation of catalyst structure, morphology, composition and support, and their influence on catalytic activity of ORR and overall performance of PEMFC, were discussed. Thirdly, the effects of optimizing the structure of the catalytic layer (CL) on the performance of MEA were analyzed. Finally, the challenges and prospects of Pt-based catalysts and catalytic layer design were discussed.

## 1. Introduction

Fossil fuels and massive energy consumption, as well as the consequent serious climate and environmental issues, pose a serious threat to the sustainable development of human society. In this regard, the utilization and development of clean energy has gradually spread across the globe. In addition to the common photovoltaic power stations and solar thermal power stations, researchers are slowly turning their attention to fuel cells [1,2,3,4]. Fuel cells can be divided into alkaline fuel cells (AFCs) [5,6,7], phosphoric acid fuel cells (PAFCs) [8,9,10], solid oxide fuel cells (SOFCs) [11,12,13], molten carbonate fuel cells (MCFCs) [14,15,16] and proton exchange membrane fuel cells (PEMFCs) [17,18,19] according to different electrolytes. In addition to the general characteristics of fuel cells, PEMFCs also have the outstanding advantages of fast response, low operating temperature, high energy conversion efficiency and high power density. A PEMFC is an energy conversion device that converts hydrogen energy into electric energy effectively through electrochemical reactions, and which can effectively alleviate the problems of global warming and fossil energy shortage. However, all the commercial PEMFC catalysts used are Pt-based catalysts, which are expensive; the stability and activity of the catalysts need to be improved, and the hydrophobic/hydrophilic balance in the catalytic layer (CL) greatly limits their application. It is of great significance to improve the overall performance of PEMFCs by improving the stability and activity of catalysts, minimizing the Pt load, optimizing the structure of each functional layer, and reducing the transmission resistance between the CL and the proton exchange membrane (PEM).

In order to boost the performance of Pt-based catalysts, a lot of research has been conducted, including on the design of the catalyst microstructure to enhance the intrinsic activity, the development of various sizes and adjustable compositions of Pt-based nanostructures, the optimization of catalyst support, etc. [20,21]. By designing the structure of a Pt-based catalyst, it can be generated on a special surface structure, so as to boost the stability and activity of Pt-based catalysts, such as low-dimensional nanostructures (e.g., two-dimensional nanoplates and one-dimensional nanowires). These structures not only have high conductivity, but can also make sure the nanocrystals are fully in contact with the support and effectively inhibit Ostwald ripening, with excellent stability [22,23,24]. The same is true of porous nanostructures, which not only provide rich catalytic active sites, but also effectively inhibit agglomeration [25]. In addition to changing the structure of Pt-based catalysts, morphology control is also an effective method to enhance the catalytic performance [26,27,28]. By generating Pt-based nanoparticles (NPs) in different morphologies, such as octahedrons or cubes, the surface with high activity can be selectively exposed [29,30,31]. Different crystal faces have different electrochemical properties. Therefore, the selective synthesis of Pt-based nanocatalysts with special morphology and specific crystal faces to improve the catalytic activity has also become a research area [32]. Alloying with other metals can also effectively improve the activities of Pt-based catalysts. After alloying with other metals, not only can the amount of Pt be reduced, but also the stabilities and activities of catalysts can be effectively improved [32,33,34,35]. At present, Pt-based alloy catalysts have also been widely used in commercial applications. For example, Toyota Mirai has used a Pt_3_Co alloy catalyst for commercial PEMFC vehicles, and General Motors has also actively tested Pt_3_Ni alloy catalysts [36]. Compared with the current commercial platinum on carbon (Pt/C) catalyst, the catalytic performance and stability have been significantly improved. In addition, the development of catalyst supports with high conductivity, high specific surface area, appropriate porosity and high stability in the PEMFC environment will also affect the catalytic performance of PEMFCs. For example, noncarbon metal oxide Ti_0.7_MO_0.3_O_2_ [37], Sn-doped In_2_O_3_ NPs [38], transition metal nitride [39], graphene-based materials [40], and metal-organic frameworks (MOFs) [41] are all effective supports for Pt-based catalysts.

The performance of PEMFCs depends not only on the catalytic activity of the catalyst but also on the structure of MEA, the core component of PEMFC to a large extent [42]. In MEA, the transport rates of electrons, gases and ions are affected by Pt/C, void space and the Nafion ionomer, respectively. Improper design can result in insufficient use of Pt-based NPs, resulting in waste of catalysts [43]. The U.S. Department of Energy (DOE), regarding PEMFC technology, has proposed to achieve Pt loading of no more than 0.1 mg_pt_·cm^−2^ by 2030 [44]. However, the Pt consumption in current conventional MEA is generally 0.2–0.5 mg_pt_·cm^−2^, and the performance of PEMFCs cannot be fully guaranteed [26,45]. In addition, the corrosion of the carbon supports will lead to the shedding and aggregation of Pt NPs, which will limit the lifetime of the catalyst layer [46,47,48,49]. At present, the development direction of MEA is long-term operation and low Pt load, and the key to overcome these problems depends on the design of the catalyst layer structure [45,50]. Therefore, in order to achieve long-term operation and low Pt load of MEA, the CL structure has been studied and explored. By designing a reasonable CL structure, the transmission and diffusion of gas, water, electrons and protons can be improved, thus improving the power density and durability of the MEA [51].

This review summarizes the recent progress in the controllable preparation of Pt-based ORR catalysts and structural design of CL, as shown in Figure 1. First, we introduce the design principle of Pt-based catalysts, analyzing the unique advantages of Pt-based catalysts in the field of ORR and the relationship between the three effects (including ligand effect, strain effect and synergistic effect) and ORR activity. On this basis, various practical methods for realizing these effects are discussed, including structural design, morphology control, compositional regulation and supports optimization, with emphasis on the influence of these methods on ORR performance and MEA performance. Then, the influence of gradient and ordered CL structure on the performance of MEA are analyzed. Finally, the challenges faced by Pt-based catalysts and CL design and future research directions are proposed.

## 2. Design Principle of Pt-Based Catalysts

The ORR on the cathode of a PEMFC is a complex reaction process involving multiple elemental reactions. In the whole reaction process, there are multiple reaction pathways (such as direct reaction pathway and continuous reaction pathway) and reactions of various intermediates (such as O_2_^2−^, O^2−^, H_2_O_2_, surface Pt-O and Pt-OH). The specific reaction mechanisms are still unclear. However, through many researchers’ theoretical and experimental studies on ORR, a practical and significant theoretical basis has been obtained for the mechanism of ORR. At present, researchers generally believe that ORR mainly takes two paths [52,53]. One is a direct four-electron pathway, in which O_2_ reacts directly into H_2_O under the action of catalyst; the reaction mechanism equation is as follows: [O_2_ + 4H^+^ + 4e^−^→2H_2_O, E^0^ = 1.229 V]. The other is the indirect two-electron pathway; under the action of the catalyst, O_2_ is first converted into the intermediate product H_2_O_2_, and then continues to obtain the electron reaction to generate H_2_O; the reaction mechanism equation is as follows: [O_2_ + 2H^+^ + 2e^−^→H_2_O, E^0^ = 0.67 V], [H_2_O_2_ + 2H^+^ + 2e^−^→H_2_O, E^0^ = 1.77 V].

Theoretical and experimental studies have found that in either four- or two-electronic pathways, the chemical bond of oxygen in the initial reaction will be broken. As a result, the reduction potential of the four-electronic pathway is higher than the two-electronic pathway, and the dissociation of the O-O bonds in O_2_ is higher than that in H_2_O_2_, thus the two-electron pathway with low catalytic activity tends to occur. However, the actual reaction is much more complicated, and the oxidation reaction at the cathode may be a mixture of two and four electrons. For PEMFC, the operation of the electric potential and current efficiency are higher when the electrode reaction is carried out in a four-electron pathway, which can avoid the influence of H_2_O_2_ produced by the indirect two-electron pathway on the proton membrane. Therefore, the four-electron pathway is also considered to be a more effective ORR process [52]. In the theoretical design and controllable preparation of ORR catalysts, the catalysts with the four-electron pathway for cathodic oxidation are more likely to be explored.

In PEMFC, the main obstacle limiting its discharge performance is the slow ORR on the cathode. The reaction cannot meet the discharge requirements of PEMFC without external force, but can be carried out with the ORR smoothly under the action of appropriate catalysts. Pt has been used as an electrocatalyst for PEMFC for decades due to its unique properties of accelerating ORR reactions [54]. Compared with other single-element catalysts (Ag, Pd, Au, Cu, Co and Ni, etc.), Pt has special electronic properties and has moderate interactions with oxygen-containing substances, so it is more favorable to catalyze ORR [55]. The ORR activity can also be reflected according to the binding capacity of single element catalyst with O or OH, as shown in the “volcano” diagram in Figure 2a [56].

Over the past two decades, numerous explorations and experiments have been conducted to prepare Pt-based catalysts with high activity, high stability and low cost, which are partially summarized in Table 1. According to Sabatier’s principle, the interaction between the surface atoms of the catalyst and the reactants should not be too strong or too weak [57]. Therefore, in order to prepare high performance Pt-based catalysts, it is necessary to adjust the binding ability of catalysts with oxygen by optimizing the electronic structure of the Pt-based surface. Many theoretical mechanisms, including strain effects (e.g., tensile or compressive strains on the surface of Pt atoms), ligand effects (e.g., electron transfer between Pt atoms and other metal atoms), and synergistic effects (e.g., interactions between Pt atoms and supports) have been shown to regulate electron structures effectively.

In practical applications, strain effect and ligand effect often act synergically on Pt-based catalysts. Strain effect enhances the performance of Pt-based catalysts by adjusting the distance between Pt atoms to change their electronic structure, as shown in Figure 2b. Ligand effects usually occur on different atomic surfaces with different d-band centers, which contribute to the transfer of electrons or charges between them [58,59], as shown in Figure 2c. For example, after alloying Pt with transition metal Ni, the center position of the d-band of Pt_3_Ni (111) shifted negatively by 0.34 eV, and its surface ORR activity was 10-times higher than that of the corresponding Pt (111) surface [60]. In addition, the ligand effect can effectively reduce the difference of electronegativity and mean the whole catalyst is in the lowest energy state, thus developing a novel approach to boost the catalytic performance [57]. Synergistic effects often occur in the interface construction between Pt-based nanomaterials and other components, such as Pt and metal, Pt and support, and Pt and other compounds [61]. The building of these interfaces can generate synergies between the different components to enhance electrocatalytic properties. Among them, the solid interface formed by Pt and supports can effectively relieve the agglomeration of Pt-based NPs and increase the electrical durability of catalysts. The introduction of catalyst supports can also prevent the agglomeration of Pt NPs during the operation of PEMFC. In addition, Pt NPs sometimes exhibit beneficial binding characteristics with the supports, known as the strong metal support interaction (SMSI) effect, as shown in Figure 2d. The SMSI effect tends to occur on metal and reducible oxide supports, such as In_2_O_3_, WO_3_ or TiO_2_ [62,63,64]. Compared with carbon-loaded Pt NPs, metal-oxide-loaded Pt NPs have higher activity.

## 3. Controllable Preparation of Pt-Based ORR Catalysts

So far, the studies on the stability and activity of Pt-based catalysts have mainly looked at the following four aspects:(1)The development and design of novel Pt-based nanocatalysts with more stable structure and more efficient performance, such as porous nanowires [77], hollow nanostructures [78] and highly open nanoframes [79], as shown in Figure 3a.(2)The activity of a Pt-based catalyst is closely related to the crystal face structure. The crystal face exposed by the Pt-based catalyst determines the atomic arrangement and electronic structure of the surface, which directly affects the electrochemical adsorption and decomposition of reactant molecules in the catalytic reaction process [80]. Therefore, controlling Pt-based NPs with different crystal faces or different geometric morphologies is an effective way to explore high-performance Pt-based catalysts [81], as shown in Figure 3b.(3)To enhance the intrinsic activity of Pt (increase the unit activity of Pt), the way to change the intrinsic activity of Pt itself is usually alloying, by introducing another or more transition metal elements, such as Cu, Cr, Co, Ni, Fe and other metal elements, to build a Pt-based alloy, heterogeneous structure or core-shell type [82,83]. At present, it has also become a hotspot in the research field of Pt-based catalysts [21,84,85], as shown in Figure 3c.(4)In recent years, more and more new support materials have attracted the attention of researchers, such as composite materials or ceramic materials with better stability than carbon materials. The physicochemical properties and structure of the surface of such support materials play an important role in the stability and activity of the final catalyst. Moreover, the interactions between the Pt-based NPs and the supports help to regulate the electronic structure of Pt, thus further enhancing the catalytic performance of Pt-based NPs [86], as shown in Figure 3d.

### 3.1. Structural Control of Pt-Based Catalysts

In commercial applications, Pt/C catalysts are commonly used in the cathodes of PEMFCs, as they have 2–3-times more catalytic activity than that of pure Pt, but this is far from expected. The Pt/C catalyst still has obvious disadvantages [88], such as high cost, easy CO poisoning, poor stability and other problems. In order to further enhance the stability and catalytic activity of Pt-based catalysts, researchers have developed Pt-based NPs with special structures, which typically have more than 10-times higher ORR mass activity than Pt [32]. At present, Pt-based catalysts can be divided into one-dimensional (1D) nanowires/nanotubes, two-dimensional (2D) nanoplates, three-dimensional (3D) polyhedrons, nanoframes and other special structures according to their different structures [87].

Low-dimensional Pt-based nanomaterials (e.g., nanowires and nanoplates) have been widely used in electrocatalysis and energy conversion due to their special atomic arrangement and electronic structure as well as good solubility and aggregation [89]. Results show that the combination of the porous metal structure and one-dimensional nanostructure is an effective method to enhance the stability and activity of Pt-based catalysts [76,90]. A porous metal structure is a very efficient structure which can provide high specific surface area and abundant catalytic active sites. The porous structure not only enhances the efficiency of Pt, but also inherits the high conductivity of the Pt, thus maximizing the electron transfer rate in the porous nanostructures [24,70]. However, the porous metal structures tend to be less stable, often in the form of NPs, which can easily agglomerate or detach from the supports [91,92,93]. The anisotropic structure of 1D nanowires can ensure sufficient contact with nanocrystals and support, which is conducive to inhibiting the agglomeration, separation and Ostwald ripening of the nanocrystals in the reaction process, thus improving the stability [94].

At present, researchers have also prepared a series of Pt alloy catalysts with porous nanostructures, which have showed good catalytic activity [70]. Li et al. [75] developed porous Pt-Fe alloy nanowires through the dealloying method, which has a unique nanostructure. This nanostructure was composed of long and porous metal wires, and each metal wire contained 2–5 nm ligaments and pores. When the diameter of 1D Pt-based catalysts is reduced to a few nanometers or layers of atoms, Pt will largely expose surface atoms, which determine its higher activity and better durability. This is of great significance for improving ORR reactions. In addition, Cao et al. [70] synthesized composition-controllable Pt-Cu porous nanowires (PNWs) by ultrasound-assisted electric displacement reaction. By adjusting the composition, the surface strain, porous structure and electronic properties, the Pt-Cu PNWs were optimized, thus improving the ORR activity. The TEM images of Pt_0.5_Cu_0.5_ PNWs displayed that the nanowires interweave with each other to form a porous network with good flexibility, as shown in Figure 4a,b. Nanowires were composed of many randomly oriented small NPs that connected to each other to form nanopores. These open structures facilitated the proton transfer in ORR, provided abundant active sites, and increased specific surface area [95]. The single-cell test showed that Pt_0.5_Cu_0.5_ PNWs had excellent catalytic activity, and their mass activity could reach 0.80 A·mg_pt_^−1^, which was about 5-times that of commercial Pt/C catalysts; its stability was also much better than that of commercial catalysts [70], as shown in Figure 4c,d. In order to further verify the performance of synthesized catalysts in PEMFC, MEA with Johnson Matthey (JM) Pt/C and Pt_0.5_Cu_0.5_ PNWs as cathode materials were measured on PEMFC pile, respectively. As shown in Figure 4e, the peak power densities of JM Pt/C and Pt_0.5_Cu_0.5_ PNWs were 398 and 476 mA·cm^−2^, respectively; the current density of Pt_0.5_Cu_0.5_ PNWs at 0.85 V was 105 mA·cm^−2^, which was 2.5-times that of JM Pt/C (42 mA·cm^−2^). The results confirmed that the ORR activity of Pt_0.5_Cu_0.5_ PNWs was improved, which was identified with the results of the single-cell tests.

In addition to the above structures, Pt-based nanoframes [24], hollow nanocages [96], single-atom Pt electrocatalysts [97] and other structures also show excellent ORR activity. Yang et al. [73] successfully transformed a PtNi_3_ polyhedron structure into a 3D Pt_3_Ni nanoframe using erosion-induced synthesis, as shown in Figure 4f. The Pt_3_Ni nanoframe structure had better stability and activity because of the change of electronic structures on the Pt_3_Ni alloy. After a long period of potential cycles, the structure of the nanoframe remained basically unchanged, and its catalytic activity hardly degraded (Figure 4g). Xia et al. [96] also synthesized a 1D bunched platinum-nickel (Pt-Ni) alloy nanocage electrocatalyst, as shown in Figure 4h. Nanocages are a kind of nano-hollow structure, which can maximize the advantages and enhance the electrocatalytic performance of Pt. In addition, the anisotropy of the 1D nanostructures enable them to have a larger surface contact with the carbon supports, leading to a higher stability. Compared with conventional catalysts, 1D bunched Pt-Ni alloy nanocages had high surface specific activity (5.16 mA·cm^−2^) and mass specific activity (3.52 A·mg_Pt_^−1^), 14- to 17-times higher than commercial Pt/C catalysts. After 50,000 prolonged durability tests, the mass activity and electrochemically active surface area (ECSA) only decreased by 1.3% and 1.1% compared with those before the test, which also shows that the catalyst has good stability, as shown in Figure 4i. Theoretical calculations and experimental results indicated that there are fewer strongly bonded platinum-oxygen (Pt-O) sites due to the ligand and strain effects. In addition, PEMFCs assembled by the catalyst also perform well. Assembled H_2_/air PEMFC provide a current density of 1.5 A·cm^−2^ at 0.6 V (Figure 4j) and can operate stably for 180 h at a constant voltage of 0.6 V. Achieving high catalytic performance with as low Pt loading as possible is the key to reducing the cost of PEMFC.

It can be seen from the above catalysts that different types of nanostructured catalysts (such as nanowires, nanocages, polyhedral nanostructures, etc.) can expose more active crystal faces and active sites to different degrees, thus improving the utilization rate and catalytic activity of Pt NPs. At the same time, this also proved that the structural control of Pt-based catalysts is of great significance for the rational design of high durability catalysts.

### 3.2. Morphology Control of Pt-Based Catalysts

In the early stages of Pt-based catalyst research, researchers focused on the composition of Pt-based catalysts, and improved the performance of Pt-based catalysts by introducing different transition metals into Pt-based NPs [30]. However, with the development of nanotechnology, the preparation of Pt-based catalysts with controllable morphology has attracted more and more interest from researchers, because the morphology of nanomaterials has a great influence on their properties, even a decisive influence.

For Pt-based ORR catalysts, the preparation of different morphologies is mainly to expose more highly active sites and thus improve the catalytic activity. In addition, Pt atoms on the surface of Pt nanocrystals of different morphologies are also arranged in different ways. For example, a Pt octahedron is dominated by Pt (111), a Pt cube is dominated by Pt (100), and a Pt concave cube is dominated by Pt (110). Common pure Pt polyhedral NPs are shown in Figure 5 [98]. Due to the different catalytic activity of Pt atoms on different crystal faces, the order of ORR activity of Pt atoms on different crystal faces is Pt (110) > Pt (111) > Pt (100) in non-adsorbent electrolytes (0.1 M HClO_4_ electrolyte), but the order of ORR activity of Pt atoms on polyhedral Pt-based catalysts is not the same, such as Pt_3_Ni catalysts. Stamenkovic et al. [99] found that Pt_3_Ni (111)-skin had much higher ORR activity than Pt_3_Ni (110)-skin and Pt_3_Ni (100)-skin, even up to 10-times of Pt (111). Considering that the ORR activity of the Pt single crystal surface was 5–10-times that of Pt/C catalyst, the activity of the Pt_3_Ni (111)-skin was about 90-times that of Pt/C catalysts. Pt_3_Ni catalysts had highly complex near-surface layers, similar to a sandwich structure. In particular, the inner and outer layers were Pt-rich, while the middle layer was Ni-rich. Surface segregation resulted in a drop in the center of the d-band, which increased weaker OH adsorption on the Pt-skin and promoted the enhancement of activity [99]. Zhang et al. [100] also found that the ORR-specific activity of Pt_3_Ni octahedral nanocrystals was 5.1-times and 6.5-times that of Pt_3_Ni cubes and Pt cubes, respectively, while the mass activity was 2.8-times and 3.6-times that of both, as shown in Figure 6, which also proved the crystal face effect of Pt_3_Ni. In addition, Wang et al. [101] prepared Pt-Ni alloy NPs by annealing after pickling, and Pt formed a Pt-skin structure on the surface of the NPs, which improved the specific activity and mass activity of Pt-Ni catalysts. The Pt-Ni octahedron prepared by Cui et al. [102] also showed good performance. Its specific activity (~3.14 mA·cm_Pt_^−2^) and mass activity (~1.45 A·mg_Pt_^−1^) were significantly higher than those of the Pt/C catalyst on the market. The improvement of this performance could be attributed to the shape of the Pt-Ni octahedron catalyst and its good surface composition.

Based on Pt crystal nucleation growth theory, the control experiment in the process of experimental conditions such as reactant concentration, reaction time and reaction temperature, etc., can effectively affect the growth of crystal nucleation thermodynamics and dynamic factors in the process of crystal nucleus to control the initial topography and Pt-accumulated growth process, and end up with a different morphology of Pt NPs. Wang et al. [66] deposited Pt cubic shells on Pd cubics to form a core-shell structure electrocatalyst. By adjusting the concentration of the Pt precursor, the transformation of Pt crystal face from (100) to (111) was realized. Due to the exposure of the Pt (111) crystal face, under the same test conditions, the mass activity and specific activity of the latter were 4-times and 3-times as those of the former, respectively. Chen et al. [103] etched the Pd-Pt nanocubes alloy using O_2_. By removing the relatively unstable Pd atoms in the outermost layer, the surface energy was reduced and the atoms migrated and rearranged. Finally, a relatively stable Pd-Pt four-dimensional (4D) nanocube structure was formed. The calculation results showed that the migration energies of Pt on Pd (111), Pd (110) and Pb (100) were 0.14, 1.72 and 1.00 eV, respectively; the migration energies of Pt on Pt (111), Pt (110) and Pt (100) were 0.28, 1.67 and 1.05 eV, respectively. Obviously, Pt atoms are easier to migrate on (111) and (100) crystal faces. At the same time, density functional theory (DFT) simulation also determined that the main crystal faces exposed by the 4D nanocubes were Pd_2_Pt (111) and Pd_2_Pt (100). The ORR activity test results showed that the MA and SA of the 4D nanocubics electrocatalyst were 11.6-times and 8.4-times higher than that of commercial Pt/C, respectively. He et al. [91] fabricated ultrathin icosahedral Pt-enriched nanocages using Pd icosahedral seeds. The catalyst had remarkable ORR activity, with 10-times and 7-times higher specific surface area and mass activity than Pt/C catalysts, respectively. The performance of the catalyst was much better than the single crystal nanocages such as cubics and octahedrals, which may be related to the dense twin defects on the catalyst surface, the exposure of the internal surface, and the Pt (111) planes being favorable for ORR.

The above studies indicate that the ORR activities of Pt-based catalysts are closely related to the Pt single crystal face. Different crystal planes have different electrochemical properties. High exponential crystal planes are composed of steps and kinks formed by low coordination atoms and have high ORR activity. Low exponential crystal planes are composed of compact atomic crystal planes and are relatively stable in thermodynamics [104,105]. Therefore, the surface structure of Pt-based catalysts can be regulated by morphology control, such as controlling the exposure of different crystal planes and preparing surface defect structures, so as to achieve the adjustment of the performance of Pt-based catalysts.

### 3.3. Composition Control of Pt-Based Catalysts

Although Pt is an ideal material for electrocatalysis, its high cost and scarcity limit the commercial application of PEMFC [21,36]. In order to improve the stability and activity of ORR, the electronic structure of Pt needs to be adjusted to solve the key problems of the surface adsorption of various oxygen-containing intermediates in the ORR process. Researchers have developed a series of Pt-based alloy catalysts for PEMFCs, which were alloyed with transition metals (e.g., Zn, Cu, Ni, Co, Fe, Mn, etc.) to change the atomic arrangement of the d-band center of Pt and the catalyst surface, thus changing the chemisorption state of oxygen-containing species on the surface of catalysts [106]. The introduction of transition metals not only effectively reduces the cost of Pt-based catalysts, but also makes their catalytic activity much higher than Pt/C catalysts due to the interaction of strain effect and ligand effect [57,83,107,108]. Mukerjee et al. [109] investigated the effect of the structure of different Pt-based binary alloy catalysts on the activity. It was found that the spacing of Pt atoms decreases after alloying, and the vacancies of Pt d-band decreased at high potential, resulting in low adsorption degree of OH_ad_ on the alloy surface. The ORR activity of Pt-M showed a “vocanic” relationship with the spacing of Pt atoms and the number of d-band vacancies, as shown in Figure 7a. That is, the spacing of Pt atoms was moderate, and the higher the number of d-band vacancies, the higher the ORR activity. Over the decades, most of the elements in the periodic table were alloyed with Pt, and the optimal combinations for high PEMFC activity have been proposed [110].

In most studies, Pt-based alloys are usually disordered structures with Pt and transition metals randomly distributed in NPs. However, acidic environments (pH < 1) and high chemical potentials (>0.7 V_RHE_) can lead to the dissolution and oxidation of transition metals, thus reducing the overall activity of the catalyst [110,111]. Therefore, the ordered Pt-based alloy catalysts, such as Pt_3_Co [74,112], Pt_3_Fe [113,114], Pt_3_Cu [115,116], etc., have attracted more and more attention. Compared to the disordered structure, the ordered Pt-based alloy catalyst has a high degree of alloying, definite composition and structure, and electronic properties, so this type of Pt alloy generally has higher electrochemical stability. Wang et al. [74] systematically studied the disordered and ordered structures of Pt-Co alloy catalysts, and synthesized ordered Pt_3_Co/C-700 NPs from disordered Pt_3_Co/C-400 alloy NPs through high-temperature heat treatment (700 °C). After 5000 potential cycles, the negative shift of half-wave potential of ordered Pt_3_Co/C-700 catalysts was less than 10 mV. Compared with the disordered Pt_3_Co/C-400 NPs, the ordered Pt_3_Co/C-700 NPs showed stronger durability. This excellent durability was mainly due to the strong interaction between Pt and Co. In addition, the mass activity of ordered Pt_3_Co/C-700 NPs was more than twice that of disordered Pt_3_Co/C-400 NPs, and specific activity increased more than threefold. The ordered formation of Pt-based alloy catalysts leads to the strong interaction between the internal components of metal, which provides an effective method for the design of efficient and durable Pt-based ORR catalysts.

In addition to the ordering process, researchers have also developed a new type of catalyst—Pt-based core-shell catalyst [117,118,119]. This catalyst can solve the previous problems effectively, requiring only a tiny amount of Pt to cover the NP surface and protect the transition metals that are easily soluble on the NP surface. The core-shell type catalyst design is very effective, because the electrochemical reaction occurs only on the surface layer of the NPs [110]. Zhao et al. [120] successfully extended high-indexed Pt alloys into core-shell nanostructures through reasonable design, forming 1D core-shell Pt-based nanowires with high catalytic stability and activity, as shown in Figure 7b. As shown in Figure 7c–e, the test results showed that the mass specific activity of Pd@PtNi nanowires with a high-indexed alloy shell was 10-times higher than that of commercial Pt/C (0.17A·mg_pt_^−1^), and the specific activity was 12-times higher than that of commercial Pt/C (0.27 mA·cm^−2^). Pd@PtNi nanowires have a unique nanostructure. The clever combination of a high-indexed PtNi alloy shell and a 1D linear core-shell structure can effectively enhance the activity of ORR. Pd@PtNi core-shell nanowires not only effectively reduce the amount of Pt metals through alloying and core-shell structure, but also improve their catalytic activity through strain effect. Moreover, the high exponential Pt-based crystal face itself has high catalytic activity, and the 1D linear structure also better enhances the stability of the catalyst.

In addition, relevant studies have shown that the stability of bimetallic Pt-based nanocrystals can be effectively improved by introducing a third metal [121,122]. Huang et al. [72] prepared highly dispersed Pt_3_Ni octahedrons on industrial carbon black by means of an efficient one-pot approach. Meanwhile, the surface of Pt_3_Ni octahedron alloys was further doped with transition metal elements (V, Cr, Fe, Mn, Co, Mo, W, Rh). They found that the Pt_3_Ni alloy octahedral surface modified with Mo can obtain higher ORR catalytic performance, and its mass activity and specific activity were 73-times and 81-times higher than Pt/C catalyst, respectively. DFT calculations showed that in the oxidation environment, Mo atoms tend to be distributed at the vertex of the octahedral surface or near the particle edges; the formation of relatively strong Mo-Ni and Mo-Pt bonds can effectively slow down the Ni atoms in the particle dissolution rate, to enhance the catalyst stability and activity [123], as shown in Figure 7f. After 8000 cycles, the Mo-Pt_3_Ni/C catalyst can still effectively maintain the octahedral shape, and no obvious degradation of its performance was observed in Figure 7g. However, the shape of Pt_3_Ni catalyst in octahedral form usually changes into round particles or concave structure after durability tests [34,72].

Compared with Pt/C, the activity of the above Pt-M catalysts was significantly improved, mainly due to the optimization of the elemental composition and surface structure of the catalysts. The effects of transition metal M on the stability of the cathode catalysts cannot be ignored. On the one hand, M slows down the mobility of Pt on the surface of carbon supports, and the particle size of the catalyst increases after alloying, which improves the anti-coalescence ability of the catalyst. The moderate dissolution of M can form a Pt-skeleton structure on the surface of the catalyst [124], which is conducive to the improvement of activity. On the other hand, the excessive dissolution of M will lead to the disappearance of alloying advantage and reduce the catalytic activity. Due to its low reduction potential, M^x+^ generated after dissolution of M cannot be reduced by H_2_ penetrated by the anode, but can combine with sulfonic acid groups of solid polymer electrolytes to occupy the position of protons. This will increase the resistance of the membrane, increase the resistance of the catalytic layer, reduce the diffusion rate of oxygen and accelerate the degradation of the membrane, and ultimately have a negative effect on the PEMFC.

### 3.4. Optimization of Pt-Based Catalyst Supports

It is well known that the catalyst support plays a key role in regulating the activity and durability of catalysts [126,127,128]. At present, carbon black materials (such as Ketjen Black and Vulcan XC-72, etc.) have been widely used as PEMFC catalyst supports due to their advantages of high conductivity, high active area, porous structure, good stability in acidic and alkaline media, and relatively low price [49,108,129]. However, in the case of PEMFCs running for a long time, the charcoal supports easily form oxygen-containing groups on the catalyst support surface, which will reduce the conductivity of the catalyst and lead to serious sintering of catalyst NPs. In addition, the oxygen-containing groups also improve the hydrophilicity of the supports, thereby affecting the air permeability and drainage performance [130]. The presence of NPs also accelerates the corrosion of carbon supports, which increases the contact resistance and reduces the thickness of the catalyst layer, resulting in the reduction of the electrocatalytic active area of the catalyst, as well as the reduction of the catalytic activity and the performance of PEMFC [131,132,133,134]. Therefore, it is necessary to explore more stable new support materials to enhance the stability and activity of PEMFC catalysts. Different types of catalyst supports have also been explored, such as carbon nanotubes (CNT) [135], carbon nanofibers (CNF) [136], mesoporous carbon [137], graphene-based materials [138,139,140,141] and metal-organic frameworks (MOFs) [142].

Recently, active exploration and research have been carried out on graphene-based materials [40,143,144,145]. Graphene materials have unique and advanced nanostructures and properties, and their unique chemical and physical properties, such as excellent electrical conductivity, high specific surface area [89,146], high thermal conductivity [147], high mechanical strength and good chemical stability, have attracted extensive attention from researchers [148,149,150,151]. Guo et al. [152] found that the deposition of the PtFe NPs on the surface of graphene (PtFe/G) can effectively improve the electrocatalytic activity and durability, as shown in Figure 8a. The double-layer capacitance of FePt/G NPs was much higher than that of commercial Pt/C or FePt/C, demonstrating that the graphene supports had a larger specific surface area than the carbon black supports, which was very significant for improving the mass activity of the Pt-based catalysts. Additionally, the ORR tests showed that the half-wave potential of FePt/G (0.557 V) was higher than that of commercial Pt/C (0.512 V) and FePt/C (0.532 V), which confirmed that the FePt/G NPs have higher ORR performance than that of the FePt/C NPs with the same Pt load, as shown in Figure 8b. Moreover, the FePt/G NPs also showed good stability and did not deactivate (relative to Ag/AgCl) even after 10,000 cycles of 0.4 to 0.8 V, as shown in Figure 8c. These results also indicated that the graphene materials were indeed a promising support material for improving catalytic activity and durability in practical catalytic applications. Moreover, the studies have shown that the single-cell constructed by the graphene-supported Pt-based catalysts have better performance than conventional Pt/C cells [153,154,155]. In addition, Wang et al. [67] prepared octahedral PtNi/CNT catalysts with good structure by using the surfactant-assisted solvothermal method, and their specific activity and mass activity were 8.5-times and 5.5-times higher than the commercial Pt/C catalyst, respectively. In addition, the MEA cathodes prepared with octahedral PtNi/CNT catalyst showed excellent durability at high potential. The voltage and maximum power density decay at 600 mA·cm^−2^ were only 3.6% and 4.8%, respectively. The results demonstrated that the graphite-structured carbon materials have practical application potential for fabricating the octahedral catalysts. After further improvement of the preparation process by surfactant-assisted solvothermal method, the carbon-supported octahedral catalysts are expected to be used in PEMFC cathodes.

Similarly, some metal oxides are widely used in various fields due to their appropriate specific surface area, mechanical strength, and high electrochemical stability. Wang et al. [156] developed a novel support prepared from highly crystalline and dispersed Ta_2_O_5_-modified CNTs, as shown in Figure 8d. The presence of highly crystalline Ta_2_O_5_ induced the growth of polyhedral-structured Pt NPs while exposing abundant (111) and (100) faces. Thus, the ORR activity of Pt-Ta_2_O_5_/CNT was improved. The electrochemical surface area of Pt-Ta_2_O_5_/CNT was 78.4 m^2^·g^−1^ and the mass activity was 0.23 A·mg^−1^_Pt_ at 0.9 V. In addition, Pt-Ta_2_O_5_/CNT also had excellent durability. After 10,000 accelerated degradation test (ADT) cycles, the performance of the Pt-Ta_2_O_5_/CNT catalyst did not degrade significantly, and the significant durability can be attributed to the strong metal-support interaction (SMSIs) effects between Ta_2_O_5_ and Pt. This can not only boost the ORR activity by triggering the electron transfer from Pt to Ta_2_O_5_, but also enhance the stability of the catalyst by inhibiting the sintering of the catalyst, as shown in Figure 8e,f. In addition, tungsten trioxide (WO_3_), as a kind of transition metal oxide, can also replace carbon materials as the support of catalysts. It has special electrochemical characteristics and good corrosion resistance. Dou et al. [157] used silicotungstic acid as the tungsten source and mesoporous silica (SBA-15) as the hard template to prepare mesoporous WO_3_ nanoclusters using hard template method, and used them as anode catalyst supports for PEMFC. Mesoporous WO_3_ nanoclusters were composed of parallel nanorods (about 8 nm in diameter), which are highly crystalline, with specific surface of 47 m^2^·g^−1^, and which have good electrochemical stability. Pt/WO_3_ prepared by glycol reduction method showed excellent electrochemical stability at high potential. After oxidation at high potential of 1.6 V for 10 h, its ECSA decreased by only 13.8%, while Pt/C decreased by 51.0%. However, the problem of low conductivity of WO_3_ still needs to be further solved.

Catalyst supports are an important part of battery systems, which have important influence on catalyst performance, fuel and charge transfer. Firstly, the support affects the dispersity, stability and utilization of catalyst NPs, which is reflected in the size and distribution of catalyst particle size, the alloying degree of catalyst, the electrochemical active region of CL, and the life of the battery. Secondly, the support affects the mass transfer process of PEMFCs; whether the reactants can fully contact with the active sites in the CL, and the transfer of reactants and products, have an important relationship with the supports. In addition, the support conductivity is directly related to the charge transmission efficiency and speed, which will directly affect the working efficiency of the whole PEMFC system. Therefore, the research and optimization of catalytic support is also an important area to further enhance the performance of PEMFC.

## 4. Effect of CL Structure on the Performance of MEA

The MEA modules of PEMFC are composed of catalytic layers (CLs), polymer electrolyte membranes (PEMs) and gas diffusion layers (GDLs) on both sides of the membrane [158]. The electrochemical reaction of the PEMFC occurs in the MEA. Specifically, the reaction gases H_2_ and O_2_ are transferred to the CLs at the anode and cathode through the diffusion layers, respectively. H_2_ is oxidized to protons on the anode, while O_2_ will be reduced to oxygen ions on the cathode, and combine with protons transferred from the anode side to form water molecules [159], as shown in Figure 9. Although the operation of PEMFC requires water to ensure adequate electrical conductivity, the reaction can be hampered if excess liquid water is not properly treated. Therefore, researchers have attempted to improve the performance of the membrane electrode by constructing a cathodic catalytic layer (CCL) with a special structure to achieve a hydrophilic/hydrophobic balance or adding hydrophobic materials to improve its hydrophobicity [160].

The electrochemical reaction on the MEA of PEMFC is a multiphase reaction, which is carried out on the three-phase boundary (TPB) formed by electrolytes, reaction gas (oxygen/air and hydrogen) and catalysts. Because PEMFC adopts solid electrolytes, its sulfonate radical is fixed on the resin of PEM and will not invade the electrode; the TPB of the reaction is limited to the part where the reaction layer contacts the PEM. Only the surface of CL and PEM can effectively conduct protons. Therefore, in order to ensure that the reaction takes place in the electrode CL, it is necessary to establish ion channels in the CL. Moreover, since the reaction requires gas reactants to reach the catalyst surface and the generated water to leave the surface quickly, it also requires that the CL has hydrophobic channels. In addition, the ionomers in the CLs will occupy part of the pores, so it is necessary to adjust the content and distribution structure of ionomers in the CL to meet the needs of proton conduction and prevent too many ionomers from reducing the porosity of the CL and affecting the transmission of O_2_. The ideal catalytic layer should have an excellent electrochemical three-phase interfacial reaction region, allowing good transfer and transport of electrons, protons, oxygen, and water [161]. Thus, it is necessary to improve the CL structure, among which the gradient structure and the ordered structure design are very effective approaches.

### 4.1. Gradient Catalyst Layer Structure

In the CL, poly tetra fluoroethylene (PTFE) and ionic conductive polymers (such as Nafion) are commonly used as binders. The content of the binder affects the gas permeability, catalytic activity and ionic conductivity of the CL [162]. If the Nafion content is too low, the ionic conductivity of CL will be insufficient, resulting in low Pt utilization and high impedance. In contrast, too much Nafion will fill the pores in the layer, affecting the gas transport and electron conduction of the CL [163]. Therefore, Nafion content in the CL should be controllable within a reasonable range, and its distribution should be gradient design.

Several studies have shown that decreasing Nafion content along a certain gradient from the region near the membrane to the GDL can improve the performance of PEMFC, especially at high current density. Xie et al. [164] prepared gas diffusion electrodes (GDEs) with uniform distribution of 30 wt% Nafion, two other GDEs with a positive Nafion gradient distribution (Nafion content in GDEs was higher toward CL/membrane interface, GDEs with a lower orientation towards CL/GDL interface), and a negative Nafion gradient distribution (Nafion content in GDEs was lower toward CL/membrane interface and higher toward CL/GDL interface), respectively, as shown in Figure 10a,b. The experimental results showed that when the Nafion content in GDEs was in a positive gradient distribution, the cathode performance improved and the battery power reached the maximum. Electrochemical impedance spectroscopy (EIS) analysis showed that the increase of battery power was related to the high Nafion content in PEM side, because it improves the proton conductivity of CL and reduces the ionic impedance at the interface between PEM and CL. In addition, the lower Nafion content on the GDL side leads to higher porosity, which reduces the resistance to gas migration and drainage. Taylor et al. [165] also established a gradient design for catalysts in CL in order to enhance the reaction activity and reduce the Pt load, as shown in Figure 10c,d. Starting from the PEM side, Pt load on carbon black decreased successively (50 wt%/20 wt%/10 wt%). The catalytic performance of the gradient structure was better than that of the uniform distribution structure when the total Pt load was basically the same. These results showed that the performance of PEMFCs can be effectively improved with the gradient distribution of Pt, especially at high current densities, which also demonstrated that Pt can be better utilized when it is concentrated near the interfaces of PEM/CLs.

Another more effective CL gradient design is to make a gradient with both Nafion and catalyst. Su et al. [166] conducted comparative analysis of CLs with three different structures: (1) Single catalyst layer (SCL) structure: Pt loading was 0.2 mg·cm^−2^ (Pt loading of the three CLs was consistent), Nafion content was 33wt%; (2) Traditional double catalyst layer (DCL) structure: only Nafion content was gradient design (Nafion content on GDL side was 20 wt%, Nafion content on PEM side was 33 wt%), Pt load on carbon black was 40 wt%; (3) The new DCL structure: Nafion content and Pt/C catalyst were in a gradient at the same time. Nafion content on GDL side was 20 wt% and Pt load on carbon black was 10 wt%, while Nafion content on PEM side was 33 wt% and Pt load on carbon black was 40 wt%. The results show that the MEA with new DCL structure has better activities than the MEA with SCL structure or MEA with traditional DCL structure. The current density of the MEA with a new DCL structure can reach up to 1.04 A·cm^−2^ at 0.6 V, which was 35.9% higher than the MEA with SCL structure, and 24.8% higher than the MEA with traditional DCL structure. Obviously, this improvement was related to the simultaneous gradient of Nafion and Pt/C catalyst. On the one hand, the lower Nafion and catalyst content on the GDL side reduces the mass transfer resistance of CL, which is conducive to the diffusion of oxygen in CL and the discharge of product water. On the other hand, impedance analysis shows that the new two-layer structure reduces the charge transfer impedance of MEA, so it has a more effective electrochemically active layer.

In general, in the high reaction zone of ORR, increasing Nafion content and Pt load can reduce proton transfer resistance and improve electrochemical reaction activity. In the low reaction region, the high proton conductivity and catalytic activity are not required, so the content of catalyst and Nafion can be reduced, which not only enhances the use efficiency of Pt, but also reduces the mass transfer resistance of oxygen diffusion and water discharge. Although the gradient design improves the MEA performance to some extent, the distribution of pores and substances in CL is disordered, and the mass transfer overpotential is still very high, and thus needs further improvement.

### 4.2. Ordered Catalyst Layer Structure

At present, the effects of the catalyst-coated membrane (CCM) [167,168,169,170], gradient catalyst layer structure [104,171] and ordered catalyst layer structure on the performance of MEA have been studied, and some achievements have been made. CCM directly coats the catalyst ink on both sides of the PEM to form CLs, which can effectively improve the catalyst utilization rate and greatly reduce the proton transfer resistance between PEM and CL. The gradient concentration CL was designed to better balance the relationship between operating conditions and electrode structures, electrode performance and precious metal load. However, both the catalyst-coating catalytic layer and the gradient concentration design catalytic layer have shortcomings. The disorder of multiphase transport channels of electrons, protons, water, gas and others will increase the mass transfer resistance and greatly reduce the utilization efficiency of catalysts [45]. Therefore, it is of great significance to study the structure of ordered catalytic layers in order to reduce the amount of Pt, enhance the utilization efficiency of catalysts and increase the three-phase interfaces of reactions.

The concept of ordered membrane electrodes was first proposed by Middelman et al. [172], as shown in Figure 11a. In the process of CLs preparation, porous ordered structure and nanoarray structure were introduced to achieve the separation and ordering of mass transfer channels, protons and electrons. The three-phase reaction interface of the electrodes was utilized to the maximum extent, the content of precious metal Pt was effectively reduced, and the stability of the electrode structure was improved. In recent years, with the deepening of research, other ordered membrane electrodes have appeared, such as carbon nanotubes/nanowires arrays, metal oxide arrays, conductive polymer arrays as the support or Pt nanostructure arrays directly as the electrode, which have different characteristics and disadvantages. Murata et al. [173] prepared ordered carbon nanotube arrays as the support and supported 0.1 mg·cm^−2^ Pt catalyst, which showed good electrochemical performance. However, when carbon-based materials are used as the supports, two-electron processes tend to occur during the operation of PEMFC, resulting in the intermediate product H_2_O_2_. H_2_O_2_ is released faster in thin CL, which leads to membrane corrosion and degradation, thus affecting the stability of the cells.

In addition, the stability of carbon materials in strong acid, strong oxidation and high potential environments also needs to be considered. In view of the fact that carbon supports are not resistant to high potential oxidation, while metal oxides have strong oxidation and acid resistance, they can possibly replace carbon materials as catalyst supports for PEMFC, especially NbO_2_, TiO_2_, WO_3_, etc. At the same time, there are strong forces between these materials and Pt-based catalysts, which can effectively enhance the stability and activity of catalysts. Jiang et al. [174] obtained titanium nitride (TiN) arrays by nitriding TiO_2_ nano-arrays in high-temperature ammonia to improve the electronic conductivity of array supports. PtPdCo was deposited on the surface of TiN nanorod arrays by magnetron sputtering, and PtPdCo tin electrodes were prepared by calcination at medium temperature. The prepared PtPdCo-TiN electrode was tested as a single-cell cathode (at a low Pt load of 66.7 μg·cm^−2^), and its mass specific power density was 5.85 W·mg^−1^, which was better than 2.46 W·mg^−1^ of commercial GDEs. Upon comparing the initial ECSA with the ECSA after 2000 cycles of potential cycle-accelerated degradation tests, the results showed that the PtPdCo-TiN cathode retained 72.9% of the ECSA of the electrode after 2000 cycles of stability tests, while the commercial GDEs retained only 59.5%, indicating the good stability of PtPdCo-TiN cathode. Xia et al. [175] also carried out in-depth research on ordered membrane electrode, preparing polypyrrole (PPy) and Nafion copolymer, as shown in Figure 11b. Pt NPs were formed on the surface of the nanowire by in-situ self-assembly technology, forming an anchor structure. This method was widely used in 1D nanowire/nanotube structures. The charged groups commonly added are polydiallyl dimethyl ammonium chloride (PDDA), Nafion, and ionic liquids. The electrode structure showed good performance in PEMFC systems, and its mass specific power density was as high as 5.23W·mg^−1^. The peak power density of Pt-NfnPPy membrane electrode (0.065 μg_Pt_·cm^−2^) can be as high as 0.778 W·cm^−2^ in H_2_/O_2_ conditions. This was comparable to the Pt/C (E-Tek) NPs cathode with a loading capacity of 0.198 mg·cm^−2^, as shown in Figure 11c. In the H_2_/air test case, the power density was slightly higher than that of the Pt/C electrode with three times the Pt load, as shown in Figure 11d.

At present, there are still some problems in the structure of ordered CLs. For example, it is difficult to control the structure of catalysts on the conductor array. Pt NPs are usually obtained by chemical synthesis, and there is a lack of methods to finely regulate the surface structure of NPs. At the micro scale, it is difficult to maintain the 1D structure of the conductor array, especially the hot-pressing step during the construction of the membrane electrode, and it is easy to destroy the ordered structure. Large-scale production also is a huge challenge. However, the ordered CL structure is beneficial to reduce the amount of catalyst and significantly improve the utilization rate of the catalyst, which is the development direction of the next generation of membrane electrodes. In general, the structure of the catalytic layer affects the mass transfer process and the utilization rate of the catalyst, and also determines the performance of the membrane electrode. Therefore, in order to maintain the power output of the membrane electrode with high performance and stability, it is necessary to build the catalytic layer structure with excellent structure. The mass transfer process is an important process affecting chemical reactions in membrane electrodes. The construction of the catalytic layer needs to consider the quality of the mass transfer ability, and the construction of the gradient and ordered catalytic layer is an important research direction in the future.

## 5. Conclusions and Perspectives

There has been remarkable progress made in the development of PEMFCs over the last few decades. The load of precious metal Pt has decreased from above 10 mg·cm^−2^ to 0.2–0.5 mg·cm^−2^, and the cathode catalyst has gradually changed from Pt/C to high-performance Pt-based catalysts. The structure of CL has also gradually evolved from traditional structures to gradient and ordered structures. All these exciting achievements indicate that PEMFCs have good development potential. In this paper, we introduced the research progress of the controllable preparation of Pt-based ORR catalysts and the structural design of catalytic layers in recent years. We also proposed some strategies to improve the performance of Pt-based catalysts, including the optimization of Pt-based nanostructures, the control of the composition of Pt-based catalysts, the selection of exposed crystal planes to control their catalytic activity, and the improvement and optimization of support. In addition, the influence of improving the catalytic layer structure on the performance of MEA was also reviewed. By designing a reasonable catalytic layer structure, the diffusion of gas, water, electrons, and protons and the utilization rate of Pt-based catalysts can be improved, and thus the performance of PEMFA can be improved. Increasing the power density, reducing costs and improving the durability of PEMFCs are the main ways to promote the large-scale commercialization of PEMFC.

Although the construction of PEMFCs has made some progress in reducing costs and improving reliability and durability, some challenges remain to be overcome. At present, many synthesis methods have been developed to improve the performance of Pt-based catalysts, but it is still a challenging task to simultaneously enhance the durability and activity of Pt-based catalysts, because Pt-based catalysts with higher activity are usually less stable. In the future, efforts should be made to design higher performance PEMFC catalysts combining multiple strategies. In addition, most Pt-based catalysts are usually evaluated using the rotating disc electrode (RDE) method, and few practical applications have been reported at the MEA level. It is well known that good ORR activity from RDE tests does not always translate into good MEA performance because the latter is in a more complex environment. Therefore, we suggest adopting a new evaluation system to solve these problems, such as the standardized evaluation of battery testing technologies and the design of gas diffusion electrodes. The ordered structure of the MEA is beneficial to reduce the amount of catalyst, and the appropriate carrier can enhance the stability of the electrode. However, so far, the ordered structure of the membrane electrode that can be used in mass production still needs further research.

In order to further reduce the cost of PEMFCs in the future, it is necessary to design Pt-based catalysts reasonably, so that they not only have a low platinum load, but also have high stability and activity. In addition, the reasonable design of catalytic layers will help to enhance the utilization rate of catalysts and provide a suitable reaction environment for electrochemical reactions. The study of the gradient CL structure is conducive to further exploring the distribution of different reaction positions in the electrodes and understand the electrochemical reaction process. The novel electrode structures represented by the ordered membrane electrodes are a good choice and are expected to become a development direction of the membrane electrode in the future. It is believed that PEMFCs will be widely used in daily life in the near future.

## Figures and Tables

**Figure 1 nanomaterials-12-04173-f001:**
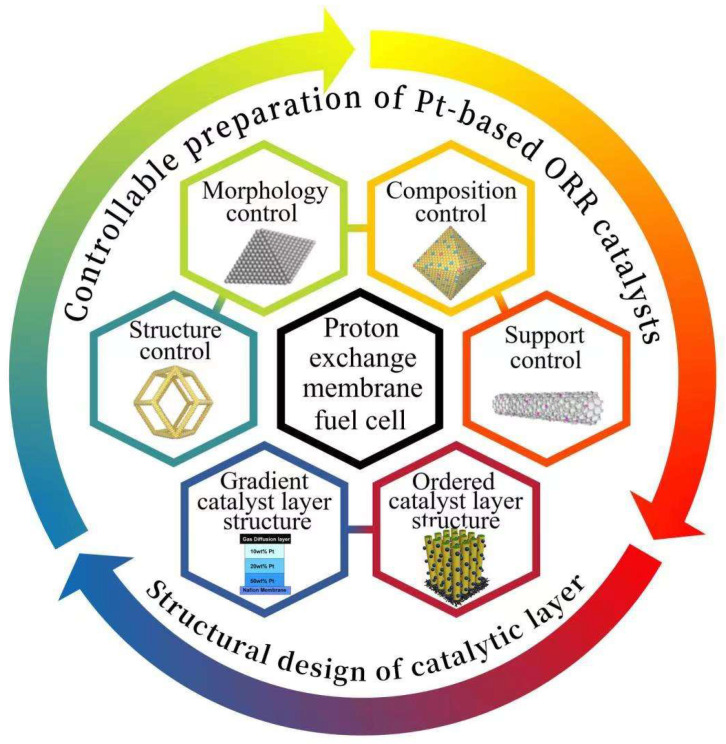
Overview framework of Pt-based oxygen reduction reaction catalysts in proton exchange membrane fuel cells.

**Figure 2 nanomaterials-12-04173-f002:**
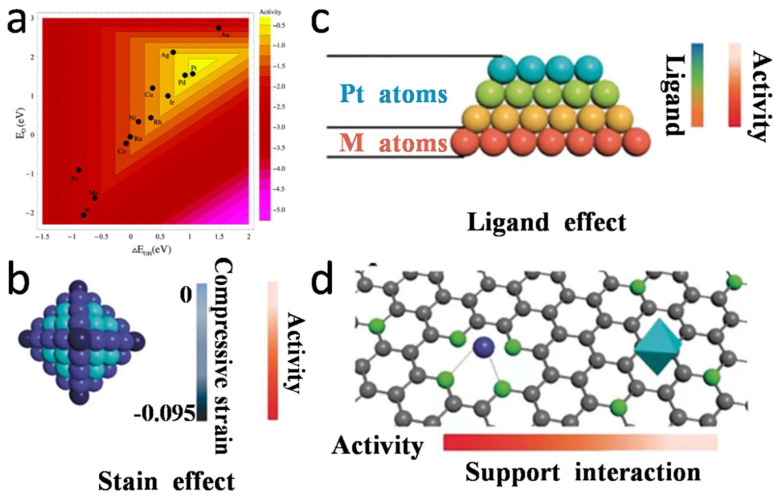
(**a**) Trend of ORR activity plotted as a function of the binding energies of the O and OH. Reproduced from Ref. [56] with permission from American Chemical Society, copyright 2004. The schematic diagram of the theoretical mechanism is as follows: (**b**) Strain effect, which refers to the surface strain distribution and the resulting relative activity; (**c**) Ligand effect, which gradually decreases with the increasing number of Pt atomic layers, almost disappearing after more than 3 layers; Reproduced from Ref. [65] with permission from American Chemical Society, copyright 2012. (**d**) Support interaction, which refers to the anchoring of a single Pt atom (blue) and the Pt NPs (octahedron) supported by heteroatom (green) doped C (gray). Reproduced from Ref. [60] with permission from Wiley-VCH, copyright 2020.

**Figure 3 nanomaterials-12-04173-f003:**
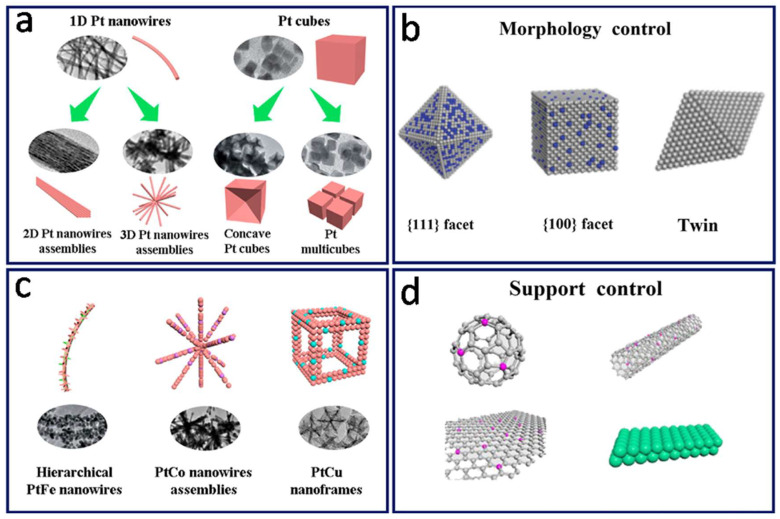
(**a**) Structure control of Pt-based catalysts. (**b**) Morphology control of Pt-based catalysts. (**c**) Composition control of Pt-based catalysts. (**d**) Optimization of Pt-based catalyst supports. Reproduced from Ref. [26] and Ref. [87] with permission from Wiley-VCH (copyright 2019) and American Chemical Society (copyright 2021), respectively.

**Figure 4 nanomaterials-12-04173-f004:**
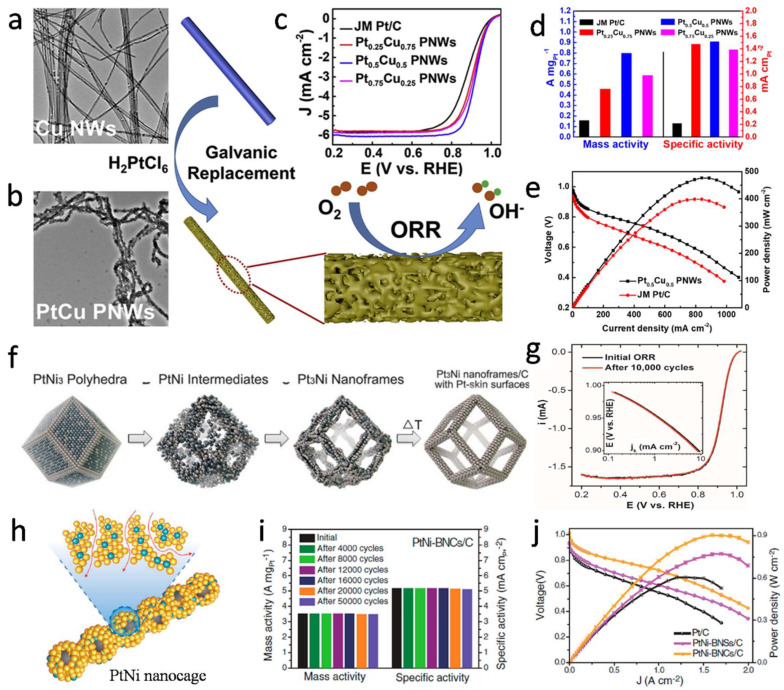
TEM images of (**a**) Cu NWs and (**b**) Pt_0.5_Cu_0.5_ PNWs; (**c**) Linear sweep voltammetry (LSV) curves (O_2_-saturated 0.1 M HClO_4_ solution at 10 mV·s^−1^ and 1600 rpm); (**d**) Catalytic performance (0.9 V) of Pt_0.75_Cu_0.25_ PNWs, Pt_0.5_Cu_0.5_ PNWs, Pt_0.25_Cu_0.75_ PNWs and JM Pt/C; (**e**) Power density curves and polarization curves of Pt_0.5_Cu_0.5_ PNWs and JM Pt/C (80 °C and 100% humidity). Reproduced from Ref. [70] with permission from Elsevier, copyright 2021. (**f**) Pt-Ni nanoframes; (**g**) Tafel plots and LSV curves of Pt_3_Ni nanoframes before and after accelerated degradation test (ADT). Reproduced from Ref. [73] with permission from Science, copyright 2014. (**h**) Bunched Pt-Ni hollow nanocages; (**i**) ADT of bunched Pt-Ni hollow nanocages for 50,000 cycles; (**j**) H_2_-air PEMFC polarization plots. Reproduced from Ref. [96] with permission from Science, copyright 2019.

**Figure 5 nanomaterials-12-04173-f005:**
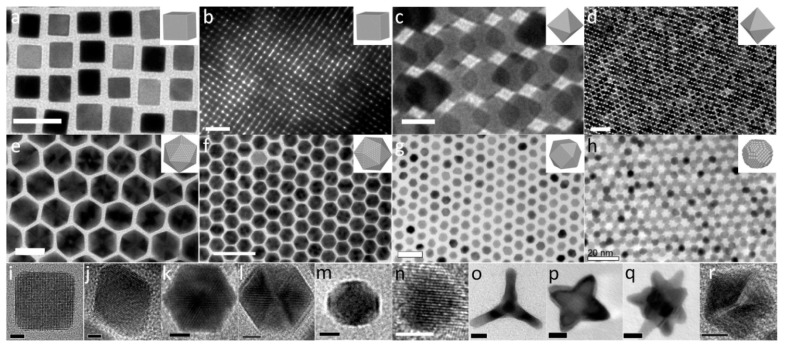
Pt NPs with different morphologies: (**a**,**b**,**i**) cubes, (**c**,**d**,**j**) octahedrons, (**e**,**f**,**k**,**l**) icosahedrons, (**g**) cuboctahedrons, (**b**,**n**) spheres, (**m**) truncated cube, (**o**) tetrapod, (**p**) star-like octapod, (**q**) multipod, (**r**) 5-fold twinned decahedron. Scale bars: (**a**,**e**,**g**,**h**) 20 nm, (**b**,**d**,**f**) 50 nm, (**c**,**o**–**q**) 10 nm, (**k**,**l**–**n**,**r**) 5 nm, (**i**,**j**) 2 nm. Reproduced from Ref. [98] with permission from American Chemical Society, copyright 2013.

**Figure 6 nanomaterials-12-04173-f006:**
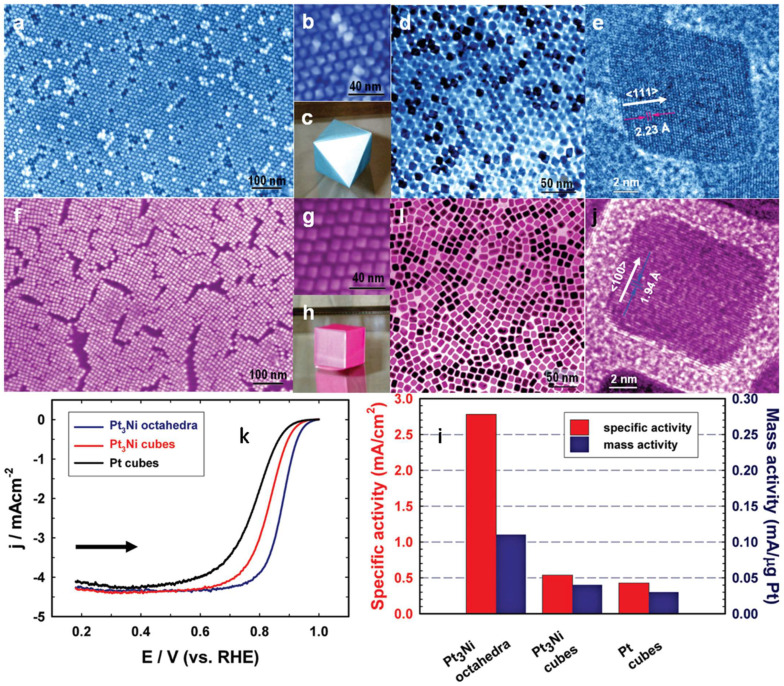
Images for (**a**–**e**) Pt_3_Ni octahedra and (**f**–**j**) Pt_3_Ni cubes: (**a**,**b**,**f**,**g**) SEM images; (**d**,**e**,**i**,**j**) TEM images; 3D image of (**c**) an octahedral and (**h**) a cube. (**k**) Polarization curves for ORR on Pt_3_Ni octahedra, Pt_3_Ni cubes, and Pt cubes; (**l**) Comparison of ORR activities on the above three catalysts. All measured at 0.9 V_RHE_ at 295 K. Reproduced from Ref. [100] with permission from American Chemical Society, copyright 2010.

**Figure 7 nanomaterials-12-04173-f007:**
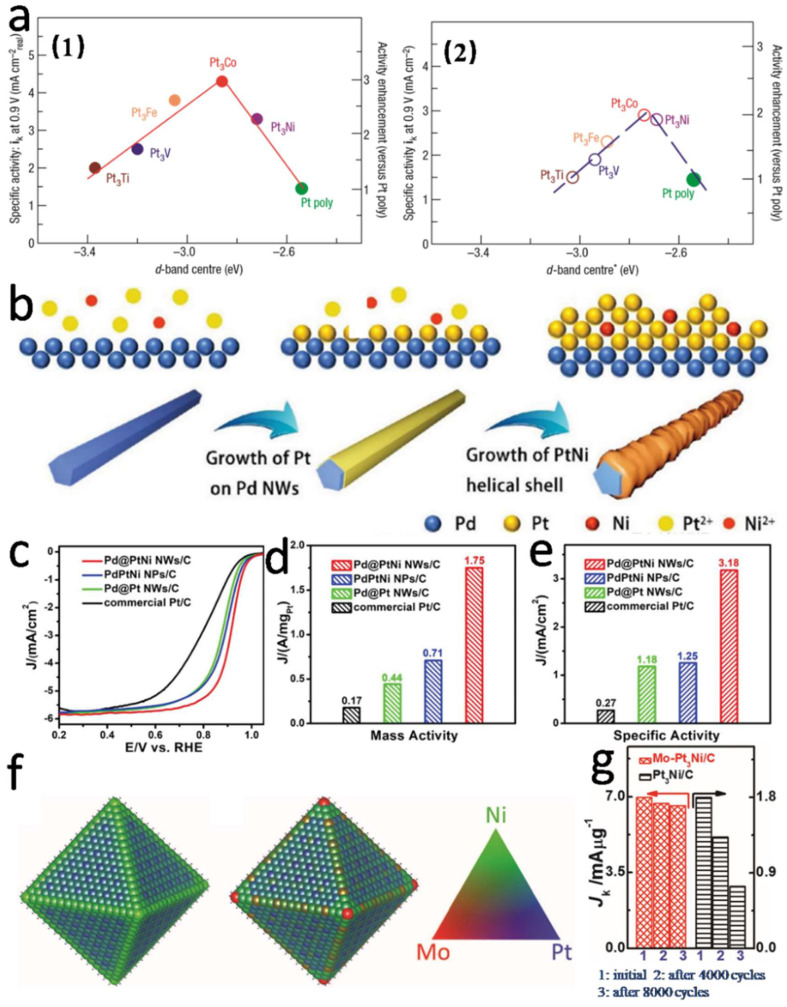
(**a**) The relationships between experimental specific activity of ORR on the Pt_3_M surface in 0.1 M HClO_4_ at 333 K and the location of d-band centers on the surfaces of Pt-skin (**a1**) and Pt-skeleton (**a2**). Reproduced from Ref. [125] with permission from Springer Nature, copyright 2007. (**b**) Schematic illustration of Pd@PtNi NWs; the ORR activities of commercial Pt/C, Pd@Pt NWs/C, PdPtNi NPs/C and Pd@PtNi NWs/C: (**c**) LSV curves, (**d**) mass activities and (**e**) specific activities at 0.9 V. Reproduced from Ref. [120] with permission from Wiley-VCH. copyright 2019. (**f**) Average site occupancies of Mo_73_Ni_1143_Pt_3357_ NC determined by Monte Carlo simulation; (**g**) catalytic activities of the Pt_3_Ni/C and Mo-Pt_3_Ni/C catalysts after ADT cycles. Reproduced from Ref. [72] with permission from Science, copyright 2015.

**Figure 8 nanomaterials-12-04173-f008:**
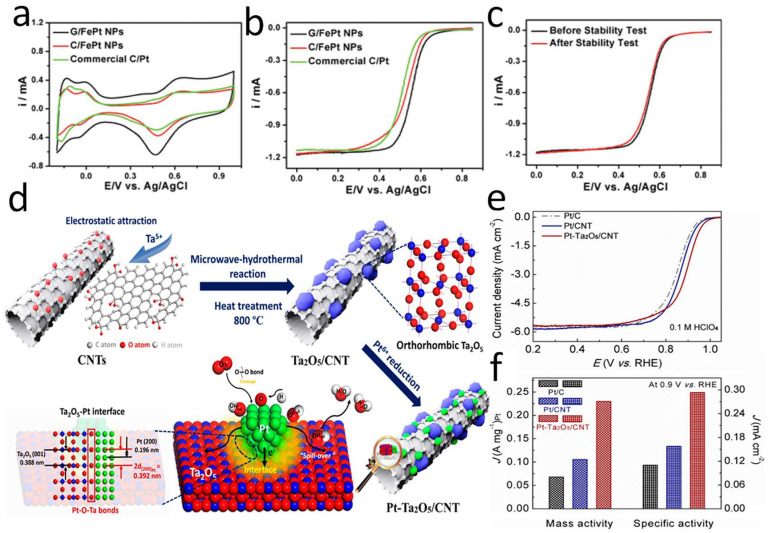
(**a**) CV curves of the obtained catalysts in N_2_-saturated 0.1 M HClO_4_ solution at 50 mV·s^−1^; (**b**) Polarization curves of the ORR catalyzed by the catalysts in O_2_-saturated 0.1 M HClO_4_ solution with a rotation speed of 1600 rpm and a potential scan rate of 10 mV·s^−1^; (**c**) The ORR polarization curves of FePt/G catalyst before and after 10,000 potential sweeps. Reproduced from Ref. [152] with permission from American Chemical Society, copyright 2012. (**d**) Fabrication process of the Pt-Ta_2_O_5_/CNT catalyst. (**e**) LSV curves of Pt-Ta_2_O_5_/CNT, Pt/CNT and Pt/C in 0.1 M HClO_4_ with a rotation speed of 1600 rpm and a potential scan rate of 10 mV·s^−1^; (**f**) Comparison of specific activities and mass activities at 0.9 V. Reproduced from Ref. [156] with permission from American Chemical Society, copyright 2019.

**Figure 9 nanomaterials-12-04173-f009:**
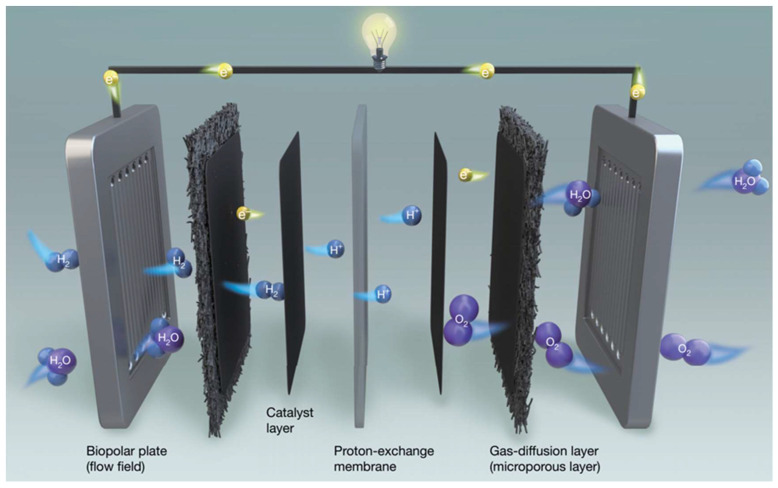
Composition of PEMFCs and electron and charge transport during operation. Reproduced from Ref. [17] with permission from Springer Nature, copyright 2021.

**Figure 10 nanomaterials-12-04173-f010:**
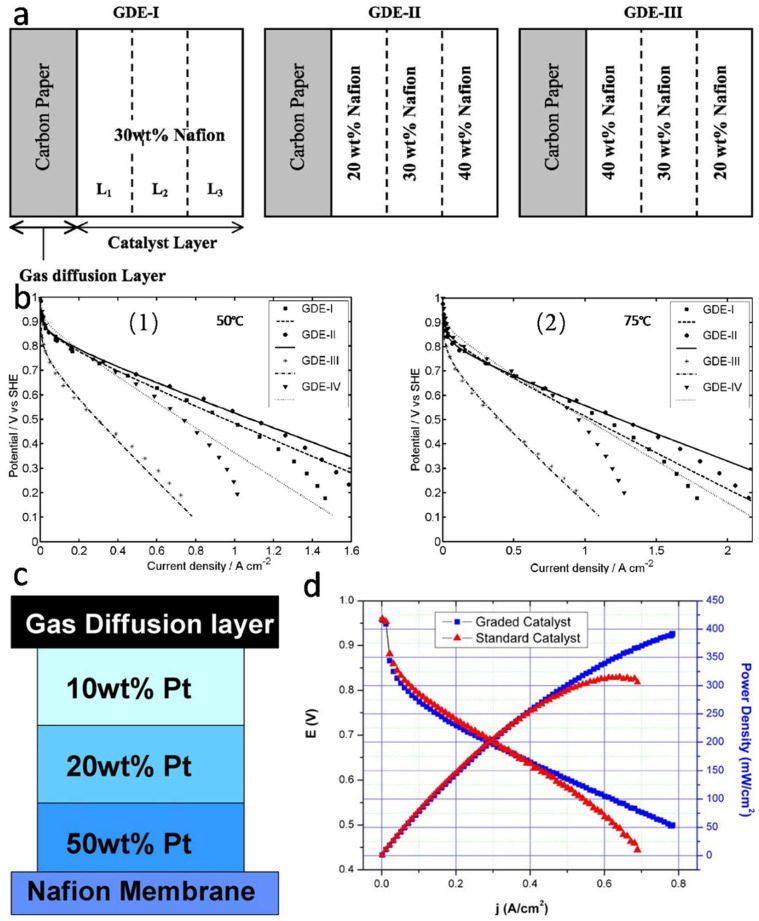
(**a**) Schematic of gas diffusion electrodes (GDEs) based on CLs with uniform and graded contents of Nafion; (**b**) Effects of the Nafion distribution in cathode on polarization plots: (**b1**) 50 °C, (**b2**) 75 °C. Reproduced from Ref. [164] with permission from The Electrochemical Society, copyright 2005. (**c**) Schematic diagram of grade-structured catalyst layers; (**d**) Performance comparison between graded catalyst and standard uniform catalyst. Reproduced from Ref. [165] with permission from Elsevier, copyright 2007.

**Figure 11 nanomaterials-12-04173-f011:**
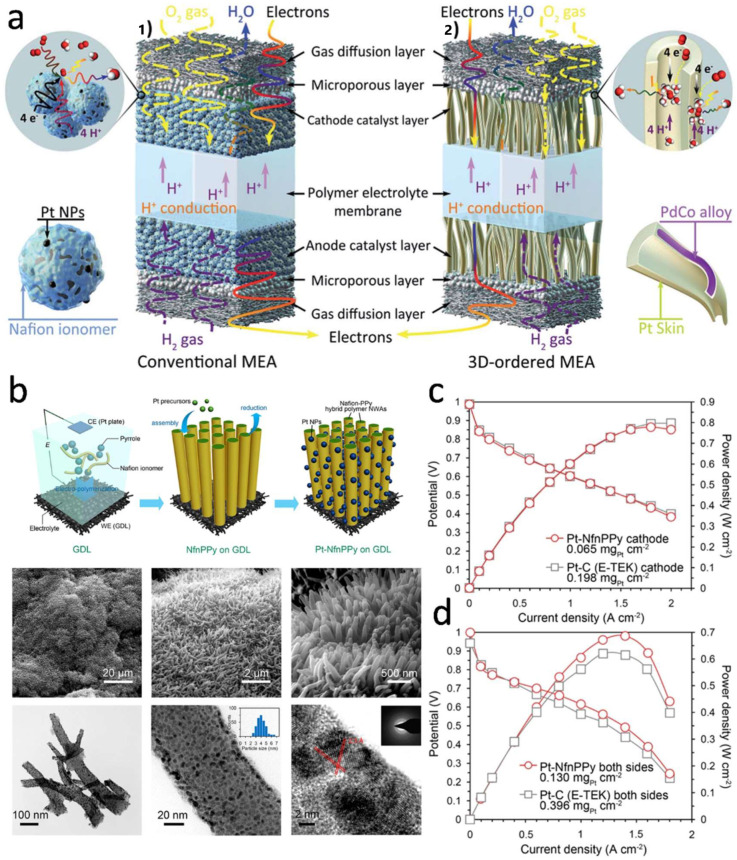
(**a**) Conceptual diagrams of (**a1**) conventional MEA and (**a2**) 3D-ordered MEA. Reproduced from Ref. [172] with permission from Royal Society of Chemistry, copyright 2018. (**b**) Schematic of the preparation and formation of Pt-NfnPPy electrode arrays; (**c**) Polarization curves in a fully humidified H_2_/O_2_ supplement at 70 °C; (**d**) Polarization curves of PEMFC equipped with Pt-NfnPPy and conventional Pt/C electrode on both sides of the MEA in a fully humidified H_2_/air supplement at 70 °C. Reproduced from Ref. [175] with permission from Springer Nature, copyright 2015.

**Table 1 nanomaterials-12-04173-t001:** The mass specific activities and surface specific activities of various Pt-based catalysts under 0.1 M HClO_4_. Values are recorded at 0.9 V vs. RHE.

Catalyst	Structure	Mass Specific Activity (A·mg_Pt_^−1^)	Surface Specific Activity (mA·cm^−2^)	Ref.
Pd@Pt	Nanocages	0.680	1.750	[66]
PtNi/CNTs	Octahedra NPs	0.479	1.376	[67]
Pt-Rh-Ni/C	Octahedral NPs	0.820	4.380	[68]
Pt-Co/C	Nano-branched	0.640	1.290	[69]
Pt-Fe/C	Nano-branched	0.470	0.920	[69]
Pt-Ni/C	Nano-branched	0.400	0.770	[69]
Pt_0.5_Cu_0.5_	Porous nanowires	0.800	1.520	[70]
Au-PtFe/C	Face-centered-tetragonal NPs	0.236	0.340	[71]
Pt_3_Ni/C	Octahedra NPs	1.800	2.200	[72]
Pt_3_Ni/C	Nanoframes	5.700	—	[73]
Mo-Pt_3_Ni/C	Octahedra NPs	6.980	8.200	[72]
Pt_3_Co/C-700	Core-shell NPs	0.520	1.100	[74]
Nanoporous Pt-Fe alloy	Nanowires	—	0.383	[75]
Jagged Pt nanowires	Nanowires	13.600	11.500	[76]

## Data Availability

Not applicable.

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
