# Peer review of "Pt-Based Oxygen Reduction Reaction Catalysts in Proton Exchange Membrane Fuel Cells: Controllable Preparation and Structural Design of Catalytic Layer"

_nanomaterials, 2022, doi:10.3390/nano12234173_

Round 1

Reviewer 1 Report

The work is devoted to the actual problem of creating an efficient air electrode in PEMFC. Despite the fact that the review considered the results of a fairly decent number of original papers, in general, there is a lack of clarity of presentation, composition, summary.

The main drawback: I would recommend the authors in the motivational part of the work to decide on what criteria they evaluate what is always called differently (Limiting discharge performance, ORR activity, intrinsic activity, power density, durability, reliability) in the review. It is absolutely impossible to compare different parameters or their absence. In my opinion, it is not enough to declare at the level of more or less. I would recommend to revise the manuscript and the respective references in framework of quantitative parameters for the comparison. Then the criteria selected by the authors will more consistently fit into the logic of presentation. This inconsistence also is shown in Fig. 1. The scheme does not consist of the criteria for the comparison. Such attempt makes the review soundless.

For example, please, expand what does “indicated improvement of catalytic activity” in ref [60] mean, and so on.

In the title for Fig. 6K: /-missed

Is it enough to compare only platinum load? Is it possible to distinguish influences f different parameters of the electrode performance? Which parameters responsible for the performance of the electrode are responsible for certain parameters of the structure, microstructure, shape of the air electrode etc.

The text clearly lacks a generalization and a clear analytical part.

Author Response

Reviewer: 1

Comments to the Author

The work is devoted to the actual problem of creating an efficient air electrode in PEMFC. Despite the fact that the review considered the results of a fairly decent number of original papers, in general, there is a lack of clarity of presentation, composition, summary.

Reply: Thank you very much for your suggestions. Your suggestions are very meaningful. First of all, this review has indeed consulted a considerable number of original papers, and summarized, analyzed and refined them. Secondly, in terms of presentation, composition and summary, there are some omissions in this paper, and there are some areas that are not clear enough. Therefore, we reorganized and summarized the article, and revised the relevant content of the article to make the content more orderly and clearer. Finally, thank you again for your valuable suggestions.

The main drawback: I would recommend the authors in the motivational part of the work to decide on what criteria they evaluate what is always called differently (Limiting discharge performance, ORR activity, intrinsic activity, power density, durability, reliability) in the review. It is absolutely impossible to compare different parameters or their absence. In my opinion, it is not enough to declare at the level of more or less. I would recommend to revise the manuscript and the respective references in framework of quantitative parameters for the comparison. Then the criteria selected by the authors will more consistently fit into the logic of presentation. This inconsistence also is shown in Fig. 1. The scheme does not consist of the criteria for the comparison. Such attempt makes the review soundless.

For example, please, expand what does “indicated improvement of catalytic activity” in ref [60] mean, and so on.

Reply: Thank you for your valuable suggestions. According to your comments, we have reviewed relevant articles and supplemented relevant comparison standards and vague parameters, so as to make the article more logical.

Changes 1:For example, after alloying Pt with transition metal Ni, the center position of the d-band of Pt3Ni (111) shifted negatively by 0.34 eV, and its surface ORR activity was 10 times higher than that of the corresponding Pt (111) surface. Changes in Page 4.

Changes 2:The single-cell test showed that Pt0.5Cu0.5 PNWs had excellent catalytic activity, and its mass activity can reach 0.80 A·mgpt-1, which was about 5 times that of commercial Pt/C catalyst, and its stability was also much better than that of commercial catalyst, as shown in Figure 4c, d. Changes in Page 7.

Changes 3:After 50000 prolonged durability tests, the mass activity and electrochemically active surface area (ECSA) only decreased by 1.3% and 1.1% compared with those before the test, which also shows that the catalyst has good stability, as shown in Figure 4i. Changes in Page 8.

Changes 4:Its specific activity (~3.14 mA·cmPt-2) and mass activity (~1.45 A·mgPt-1) were significantly higher than those of the Pt/C catalyst on the market. The improvement of this performance could be attributed to the shape of the Pt-Ni octahedron catalyst and its good surface composition. Changes in Page 9.

In the title for Fig. 6K: /-missed

Reply: Thank you for your valuable suggestions, this error has been modified.

Is it enough to compare only platinum load? Is it possible to distinguish influences f different parameters of the electrode performance? Which parameters responsible for the performance of the electrode are responsible for certain parameters of the structure, microstructure, shape of the air electrode etc.

Reply: Thanks for your suggestions. Platinum (Pt) loading is one of the most important factors in the evaluation of hydrogen fuel cell catalysts, and how to effectively achieve low Pt loading of catalysts is also the current mainstream research direction, because Pt is not only the main active substance in the catalyst, but also through reducing the load of Pt can more effectively reduce the cost and promote the commercialization process of PEMFC. The Pt-based ORR catalysts can be evaluated effectively by comparing Mass specific activity or Surface specific activity when exploring the influence of different parameters on electrode performance. In addition, we also compared the effects of other parameters on the electrode activity, such as current density, cycle number, half-wave potential and so on. The morphology and crystal structure of Pt material, as well as the structure of Pt catalytic layer, have important effects on the performance of the electrode, which is also the focus of this paper.

The text clearly lacks a generalization and a clear analytical part.

Reply: Thanks for your suggestions. There are indeed shortcomings in the generalization and analysis of this paper. For this, I have reorganized this paper and summarized and analyzed the content of this paper, which makes the content of this paper more abundant and comprehensive.

Changes 1:It can be seen from the above catalysts that different types of nanostructured catalysts (such as nanowires, nanocages, polyhedral nanostructures, etc.) can expose more active crystal faces and active sites to different degrees, thus improving the utilization rate and catalytic activity of Pt NPs. At the same time, it also proved that the structural control of Pt-based catalysts was of great significance for the rational design of high durability catalysts. Changes in Page 8.

Changes 2:In general, the structure of the catalytic layer affects the mass transfer process and the utilization rate of the catalyst, and also determines the performance of the membrane electrode. Therefore, in order to maintain the power output of the membrane electrode with high performance and stability, it is necessary to build the catalytic layer structure with excellent structure. The mass transfer process is an important process affecting chemical reactions in membrane electrodes. The construction of the catalytic layer needs to consider the quality of the mass transfer ability, and the construction of the gradient and ordered catalytic layer is an important research direction in the future. Changes in Page 21.

Reviewer 2 Report

Regarding Pt-based catalysts as cathode catalysts for PEMFCs, many important previous works are summarized, focusing on the structural design of catalyst to catalytic layer. Section 2 "Design principle of Pt-based catalyst" and Section 3 "Controllable preparation of Pt-based ORR catalyst" are well summarized including recent research works. Regarding Effect of CL structure on the performance of MEA in Section 4, it is a pity that very recent information is not included. However, the information on the Pt-based catalyst as a whole is summarized and useful for the readers, so I think it should be published in the journal as it is.

Author Response

Reviewer: 2

Comments to the Author

Regarding Pt-based catalysts as cathode catalysts for PEMFCs, many important previous works are summarized, focusing on the structural design of catalyst to catalytic layer. Section 2 "Design principle of Pt-based catalyst" and Section 3 "Controllable preparation of Pt-based ORR catalyst" are well summarized including recent research works. Regarding Effect of CL structure on the performance of MEA in Section 4, it is a pity that very recent information is not included. However, the information on the Pt-based catalyst as a whole is summarized and useful for the readers, so I think it should be published in the journal as it is.

Reply: Thanks for your comments. This paper does have some omissions about the recent research on the impact of CL structure on MEA performance. In this regard, I reorganize and summarize the fourth chapter, and add the recent research on the impact of CL structure on MEA performance to make the content more comprehensive.

Changes 1:Therefore, researchers attempted to improve the performance of the membrane electrode by constructing a cathodic catalytic layer (CCL) with a special structure to achieve a hydrophilic/hydrophobic balance or adding hydrophobic materials to im-prove its hydrophobicity [161]. Changes in Pages 16.

Changes 2:The ideal catalytic layer should have an excellent electrochemical three-phase interfacial reaction region, allowing good transfer and transport of electrons, protons, oxygen, and water [162]. Changes in Pages 17.

Changes 3:In contrast, overmuch Nafion will fill the pores in the layer, then affecting the gas transport and electron conduction of the CL [164]. Changes in Pages 17.

Changes 4:In general, the structure of the catalytic layer affects the mass transfer process and the utilization rate of the catalyst, and also determines the performance of the membrane electrode. Therefore, in order to maintain the power output of the membrane electrode with high performance and stability, it is necessary to build the catalytic layer structure with excellent structure. The mass transfer process is an important process affecting chemical reactions in membrane electrodes. The construction of the catalytic layer needs to consider the quality of the mass transfer ability, and the construction of the gradient and ordered catalytic layer is an important research direction in the future. Changes in Pages 21.